# Bending Properties of Mg Alloy Tailored Arc-Heat-Treated Blanks

**DOI:** 10.3390/ma12060977

**Published:** 2019-03-25

**Authors:** Daxin Ren, Fanyu Zeng, Liming Liu, Kunmin Zhao

**Affiliations:** 1School of Automotive Engineering, Dalian University of Technology, Dalian 116024, China; rendx@dlut.edu.cn (D.R.); s853619@mail.dlut.edu.cn (F.Z.); kmzhao@dlut.edu.cn (K.Z.); 2Key Laboratory of Liaoning Advanced Welding and Joining Technology, Dalian University of Technology, Dalian 116024, China; 3School of Materials Science and Engineering, Dalian University of Technology, Dalian 116024, China

**Keywords:** Mg alloy, tailored heat-treated blank, metal forming, electrical arc, pretreatment

## Abstract

Tailored heat-treated blank is a special kind of sheet, and the plastic forming ability can be improved. In this work, the poor room-temperature plasticity of a tailored magnesium alloy blank was address through arc heat treatment. The formability of the material was enhanced through local modification with arc pretreatment. The plasticity of the tailored arc-heat-treated blank was verified through the V-bending test. The microstructure and mechanical properties of the blank were tested, and the mechanisms underlying its improved deformability were analyzed. The bendability of the blank first increased and then decreased as heat input increased. The maximum V-bending ability of the arc-heat-treated blank increased by 88% relative to that of the untreated blank. Although springback decreased under increasing heat input, the local strength and elastic modulus of the alloy blank were equivalent to those of the base metal. This result indicated that the springback resistance of the material did not improve. The back of the blank treated under the optimal parameters comprised heat-affected zones with good plasticity. Recrystallization and grain growth occurred in the heat-affected zones. The blank exhibited reduced hardness and improved malleability. When the heat input was further increased, however, a semi-melting zone formed on the lower surface of the blank. The formation of this zone resulted in the precipitation of intermetallic compounds from the crystal phase and increased the hardness of the blank. It also decreased the plasticity and malleability of the blank.

## 1. Introduction

As the lightest metal structural material, magnesium alloy has good application prospects in the fields of rail transit, automobile, and aerospace [1]. Its weight reduction effect is crucial in reducing energy consumption and decreasing CO_2_ emissions [2]. However, magnesium alloy has a closely packed hexagonal crystal structure, and the sliding system is single at room temperature, resulting in poor plastic deformation ability. Magnesium alloy sheets are prone to cracking during plastic deformation, which is a disadvantage in the production of their complex structural parts [3].

At present, the use of hydroforming, incremental forming, and other methods can improve the plastic forming ability of light alloys [4,5]. The elongation of magnesium alloy can be enhanced at high temperature [6]. Therefore, the plastic processing of magnesium alloys under warm and hot conditions can be improved by partial or integral heating. Local heating mainly uses a laser or electromagnetic induction device to heat the key parts of sheet metal in real time during stamping. Integral heating involves heating of whole sheet metal to a certain temperature in a heating furnace before stamping, and the heated blank is stamped into a designed shape [7,8,9].

Tailored heat-treated blank (THTB) is a special kind of sheet metal. The key idea is to heat a specific area of sheet metal with a heat source depending on the requirement of plastic formation to change sheet metal’s performance distribution. This treatment can improve the local forming limit of sheet metal and enhance the robustness of the whole production process [10]. Given that the subsequent stamping process is realized under the conditions of cold formation, this technique has many advantages compared with traditional warm formation or thermal-assisted formation. Special and expensive components, such as hot forming tools or heat-resistant handling systems, are not required, and heat treatment operations can be separated locally and in time from the forming process. Laser is mainly used to fabricate plates with small size and high gradient properties. Induction and thermal conductivity can rapidly fabricate large-area plates by batch [11,12].

Arc and laser are commonly used heat sources in industrial applications. Compared with laser, arc treatment is a low-cost method with high efficiency. Meanwhile, single treatment leads to a large action area [13]. After cooling, the local properties of arc-treated materials will change [14]. By selecting the type of arc heat source and changing the arc parameters, the shape and energy distribution of the arc can be controlled. Through the above adjustments, the local performance of the material and the control of the processing area can be realized. The microstructures of the treated materials can be controlled by adjusting the arc energy density and cooling rate, which determine the plasticity and hardness of the treated materials [15]. The arc distribution in space can determine the width of single treatment [16,17]. Therefore, production of magnesium alloy THTB plate by arc treatment is feasible.

Arc treatment and arc welding are highly similar. In arc welding, softening of the heat-affected zone (HAZ) has different effects on the properties of sheet metal, but the plasticity and toughness of its softening structure have been improved to a certain extent, providing favorable conditions for plastic formation. Therefore, to improve the formability of magnesium alloy at room temperature, arc treatment was introduced into magnesium alloy stamping to prepare magnesium alloy THTBs. Specifically, AZ31 magnesium alloy was treated locally with gas tungsten arc (TIG) and then prepared to tailored heat TIG-treated blank (THTTB) after it was completely cooled to room temperature. V-shaped bending experiments were then carried out to study the bending fracture and springback of magnesium alloy sheets under different treatment conditions, and the structure and mechanical properties of magnesium alloy sheets after pretreatment were analyzed.

## 2. Materials and Experiments

In this experiment, the preparation and performance of THTTB were divided into three steps (Figure 1) as follows. Step 1: Key areas that may cause formation problems, such as cracks, wrinkles, or springback, were identified by finite element simulation of the stamping process. Step 2: In these critical areas, the microstructure of the material was changed by arc treatment, and the mechanical properties of the material were changed accordingly. When the arc moved at high speed, the heated areas would rapidly cool down to room temperature and be made into THTTB. Step 3: THTTB was transferred to a mold to perform the required stamping operations, including cutting, bending, drawing, forming, and flanging. In this experiment, the stamping operation was V-shaped bending, and the arc was processed along the curve.

The material used in this experiment was a 1.5 mm-thick AZ31 magnesium alloy sheet. The settings of AC argon arc treatment were as follows: tungsten electrode was perpendicular to the surface of the plate; the diameter of the tungsten electrode was 2.4 mm; the tip was 2 mm away from the surface of the plate; the moving speed of the electrode was 2000 mm/min; and the arc current range was 20–90 A. After arc treatment, a rectangular sample (100 mm long and 20 mm wide) was cut. The specimen was then placed on a V-shaped mold for bending test using a universal testing machine of 100 kN. The punch moved at a speed of 5 mm/min. When the sheet metal broke, the punch stopped running, and the formation limit of the sheet metal under V-bending condition was obtained. Subsequently, the springback experiment was carried out. After the punch moved to a specific position, it returned to the starting point, and the springback performance was determined by the force–displacement curve. Tensile properties of the arc treated sheet were evaluated at a rate of 1.0 mm min^−1^. Using this data, the tensile shear load of the welded joints was measured by the average value of three samples per welding condition. The microhardness tests were performed with a Vickers hardness tester with a period of 10 s and a load of 50 g. The microstructural changes before and after treatment were observed by an optical microscope and a scanning electron microscope (SEM) with energy dispersive X-ray spectroscopy (EDS).

## 3. Results and Discussion

### 3.1. THTTB Performance 

Figure 2 shows the THTTB of magnesium alloy after AZ31 arc pretreatment. After stable arc treatment, the surface was smooth, and the surface width of the single treatment area was 4 mm (Figure 2a). Carrying out multiple heat treatments in accordance with actual requirements was possible, as shown in Figure 2b. Laser is the traditional heat source for manufacturing THTTB, but the arc processing width is larger, and efficiency is higher than those of laser. THTTB can simplify the process and improve production efficiency under certain conditions. For example, when arc pretreatment is needed in a 1 m-long area, the forward speed of the arc is 2.0 m/min, taking only 0.5 min to complete. However, more than 5 min is needed when heating the plate to 300 °C with a heating furnace.

The closely packed hexagonal structure of magnesium alloy leads to its poor plasticity, and fracture is one of the most important problems in plastic formation. To investigate the influence of arc treatment on the formation properties of the sheet, the V-shaped bending test of an AZ31 magnesium alloy THTTB sheet under different arc currents (20–90 A) was carried out. The force and displacement curves are shown in Figure 3. The untreated AZ31 sheet had a maximum bending load of 528 N, which dropped rapidly after reaching the peak and cracked on the surface. The highest load of THTTB sheets under different current conditions was between 519 and 547 N, which did not change significantly compared with the base metal. The AZ31 magnesium alloy was not a heat-treated sheet; hence, the maximum bending strength after arcing did not change significantly. When the current was 60–90 A, the load reached a peak and then gradually decreased until the surface of the sheet cracked.

From the displacement, when the surface of the AZ31 sheet was broken, the head displacement was only 6.9 mm. As the arc current increased, the displacement increased as the rupture occurred. When the current reached 60 A, the displacement peaked at 12.2 mm. However, as the current further increased, the displacement gradually decreased. The displacement under 90 A was less than that under 50 A. By measuring the deformation angle (as shown in Figure 4), the same result as the displacement was obtained. The untreated AZ31 sheet broke when the deformation angle reached 52°, and the deformation angle increased with the increase in the welding current. The maximum value of 98° was obtained under 60 A, and the maximum angle of the deformation decreased when the current continued to increase.

Springback is one of the main factors that affects the precision of plastic formation. As illustrated in Figure 3, the base metal rapidly entered the plastic deformation stage after undergoing negligible elastic deformation. The springback of the sheet treated under different conditions was determined through the V-bending test. The bending load was unloaded when the displacement reached 6.5 mm. The resulting force–displacement curve is shown in Figure 5. According to displacement until the load decreased to 0, the springback of the sheet gradually decreased as welding current increased. The treated and untreated sheets exhibited large springback after V-bending. The springback performance of the magnesium alloy sheet did not considerably improve after arc pretreatment.

Treated sheets were subjected to tensile tests under typical parameters. The results are shown in Figure 6. The highest failure load of the arc-treated sheet was equivalent to that of the base metal, and the fracture positions of the joints were all located outside of the arc treatment area. Similar to arc-welded sheets, the arc-treated sheets retained their strength. The strength of the arc-welded magnesium alloy can approximate that of the base metal in the absence of weld defects. The overall elongation of the arc-treated area increased, and elongation increased by 23% when the optimal V-bending parameter was 60 A. The elastic modulus of the sheets did not considerably change under different current conditions. Springback performance is determined by elastic modulus, strength, and other factors. Al alloys, such as 6061-T6 [18], show drastic reductions in overall tensile strength. By contrast, the overall tensile strength of the arc-treated magnesium alloy negligibly changed. The minor change in overall tensile strength mainly accounts for the slight improvement in springback performance illustrated in Figure 5.

### 3.2. Microstructure

The cross-section obtained by treating the plate with different arc currents in a single pass is shown in Figure 7. As the arc current increased, the penetration depth and melt width of the treatment zone also increased. When the current increased to 90 A, the melting depth reached 1.3 mm or more. Simultaneously, given the effect of thermal expansion and contraction, the back surface of the sheet formed protrusions, and sheet deformation increased.

The bending test results showed that the formation ability was optimum when the arc current reached 60 A but decreased when the current further increased. Figure 8 shows the microstructure at different locations in the treated area under 60 A. The treated organization could be divided into four areas: (a) untreated base metal (BM) region; (b) melting zone (FZ); (c) partly melting zone (PMZ); and (d) heat-affected zone (HAZ). As shown in Figure 8b, the magnesium alloy with low melting point and high thermal conductivity underwent non-equilibrium solidification during arc treatment, and the liquid metal formed a dendritic structure under rapid solidification conditions. In Figure 8c, PMZ was formed around FZ. Although crystal grains were not melted, Mg17Al12 intermetallic compound was observed in grain boundaries and crystals according to the element analysis result of EDS (components of P1, P2, and P3 are very close to that of Mg17Al12). Given that the grain boundary had a low melting temperature and a high aluminum content, when the temperature interval was between the phase diagram liquid phase and solid phase line, the grain boundary melted to form the Mg_17_Al_12_ intermetallic compound (IMC). The compound in the crystal was also precipitated at a high temperature near the melting zone. Figure 8d shows the HAZ recrystallized and grown between PMZ and the edge of the sheet. The temperature in this zone was in the magnesium alloy annealing temperature range.

As the current increased from 60 A to 90 A, the FZ and PMZ regions also increased, and the limitation of the thickness caused the HAZ region under the PMZ to gradually decrease. When the current reached 90 A, the bottom of the sheet was basically composed of the partly melted zone structure shown in Figure 8c. To study the relationship between different tissues and properties, the hardness of the plates under different welding arc heat treatment conditions was measured. The hardness distribution along the upper surface to the lower surface of the sheet under the arc conditions of 60, 80, and 90 A is shown in Figure 9a. Under 60 A, the FZ hardness composed of dendritic structure was low, and the PMZ hardness adjacent thereto increased, which gradually decreased after entering the HAZ region. When the current increased to 80 A, the softening HAZ of the lower surface reduced. When the current reached 90 A, the lower surface of the sheet had completely formed PMZ with a relatively high hardness. Transversal hardness distribution (Figure 9b) shows that the hardness decreased from BM to HAZ, and the softening zone width by one pass reached 10 mm.

The magnesium alloy selected for the experiment was in a rolled state. Annealing was an effective way to reduce hardness and improve plasticity and toughness. Under the action of 60 A arc, a localized annealed structure formed in the lower part of the magnesium alloy sheet, which had a good effect on the improvement in the V-shaped bending ability. The hardness test effectively proved the correspondence between the annealing effect and microstructure. Figure 10 shows the hardness distribution of the lower surface of the sheet under 60 A. The region of low hardness formed in a region of 5 mm from the center of the treatment, which indicated that a single softening zone of 10 mm was obtained in a single treatment. The local heat treatment width could effectively improve the efficiency compared with the laser. After V-shaped bending, plastic deformation caused the surface hardness to increase again, which reduced the influence of the softening zone on the surface properties of the structural members.

A set of special tensile experiments were designed to determine the rules governing the changes in the mechanical properties of the FZ, PMZ, and HAZ zones after arc treatment. Given the uneven distribution of the FZ, PMZ, and HAZ zones, however, this set of tests cannot completely and truly characterize the stress–strain relationships of the test site under tension and can only be applied to compare the properties of different areas. The schematic of special tensile experiments is shown in Figure 11a. To ensure that the deformation area was located inside the FZ, PMZ, and HAZ zones during tension, 1.2 mm of material was machined and removed from the center of the tensile specimen. Only the relevant area with a thickness of 0.3 mm was retained. The force–displacement curve shows that the HAZ exhibited the largest displacement under tension, followed by the FZ. Elongation in the PMZ was only 73% of that in the heat-affected zone.

The lower surface of the sheet was identified as the stress concentration area through V-bending tests. The performance of this area determines the plasticity of the material. The low plasticity of AZ31 magnesium alloy originated from work hardening and grain refinement during rolling. Thus, surface cracking occurred upon minor deformation. The influence of arc treating is summarized in Figure 12. Dendrites were produced in the melting zone, and grains in the HAZ zone expanded. The hardness of the alloy was lower than that of the base metal, and the plasticity of the alloy correspondingly improved. As the welding current gradually increased, the sheet underwent local transition from condition 1 to condition 2, and its local deformability correspondingly improved. The maximum plasticity was observed when a large HAZ zone formed on the back of the sheet under the optimal heat input conditions. The PMZ zone had higher hardness and lower plasticity than the HAZ zone because a large amount of the intermetallic Mg17A112 compound precipitated in the PMZ zone. Numerous PMZ zone were generated on the lower surface of the sheet during the transition from condition 2 to condition 3. The generation of PMZ zones in this area decreased the overall deformation capacity of the sheet. The formation conditions of AZ31 magnesium alloy were optimized through the adjustment of parameters, such as arc current, tungsten electrode height, and arc torch moving speed. High numbers of HAZ zones formed on the lower surface of the sheet under the optimal formation conditions.

## 4. Conclusions

(1) THTTB, a magnesium alloy, can be fabricated by using an arc heat source. A softening area with a width of 10 mm can be obtained on the back of the sheet through single-pass pretreatment. Single-pass or multipass local heat treatment can change local properties by different degrees in accordance with given design requirements.

(2) V-bending capacity increased as arc current increased under experimental conditions. The maximum angle of the alloy reached 98°, which was 88% higher than that of the base metal. Deformability decreased as the current further increased. The springback performance of the alloy continuously improved as the current increased but did not considerably improve relative to that of the base metal. The tensile strength of the treated area of the THTTB sheet was equivalent to that of the base metal, and the overall elongation of the area increased after arc treatment.

(3) The arc action site consisted of the FZ, PMZ, and HAZ zones. PMZ formed around FZ. Intermetallic Mg17Al12 compounds were present at grain boundaries and in grains. HAZ surrounded PMZ. Recrystallization and grain growth occurred. Under the optimal formation parameters, numerous HAZ components formed on the lower surface of the sheet. Plasticity and deformability decreased and additional PMZ zones formed on the lower surface area of the sheet when the current was further increased.

## Figures and Tables

**Figure 1 materials-12-00977-f001:**
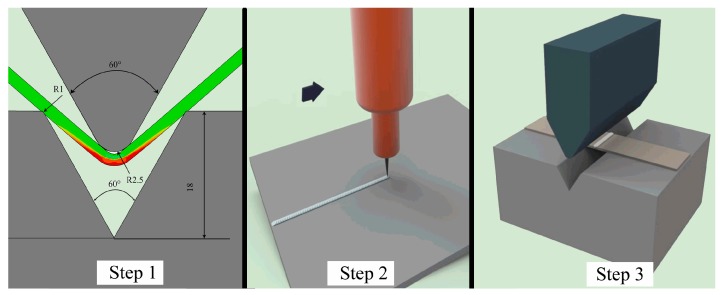
Experimental schematic diagram.

**Figure 2 materials-12-00977-f002:**
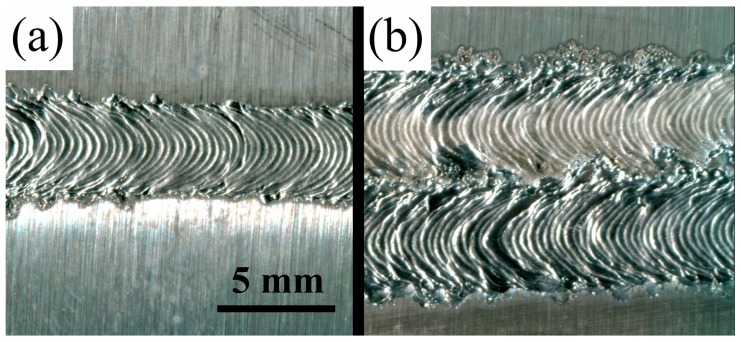
Surface morphology of tailored heat TIG-treated blank (THTTB): (**a**) Single pass; (**b**) Double pass.

**Figure 3 materials-12-00977-f003:**
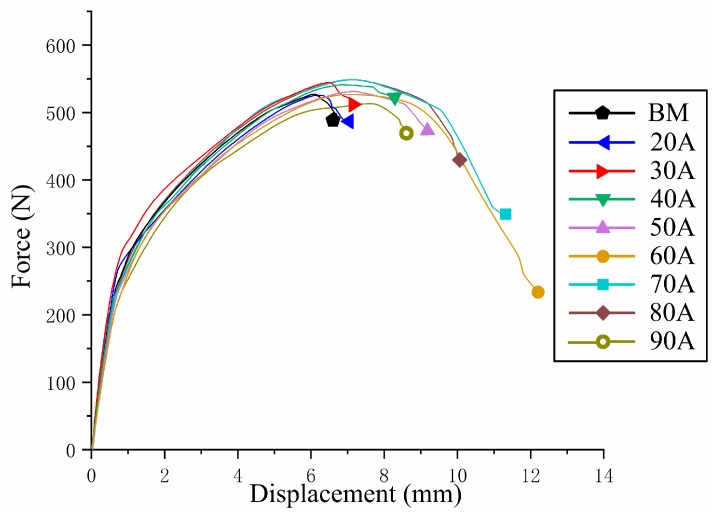
V-shaped bending force–displacement curve.

**Figure 4 materials-12-00977-f004:**
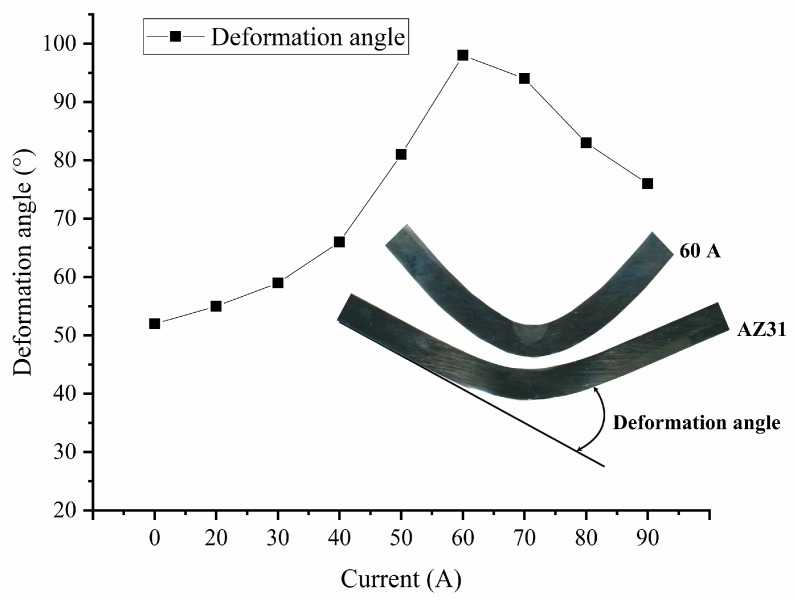
Deformation angle under different current conditions.

**Figure 5 materials-12-00977-f005:**
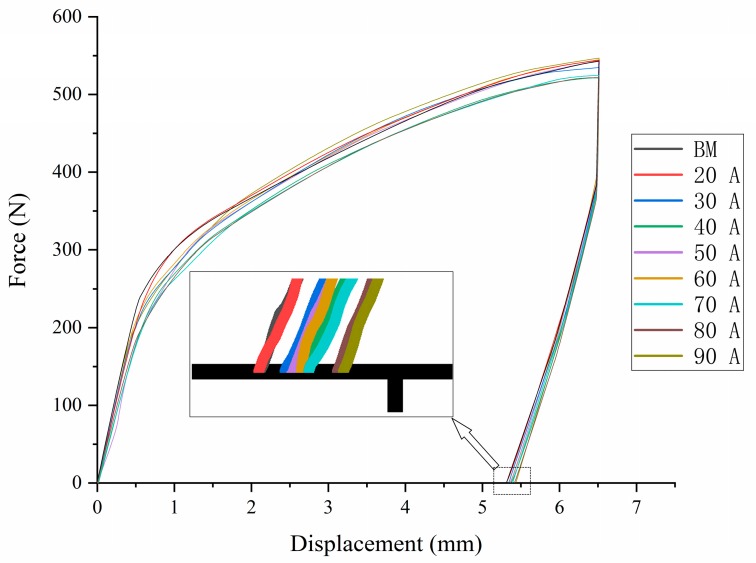
Loading and unloading force vs displacement curves of the V-shape bending.

**Figure 6 materials-12-00977-f006:**
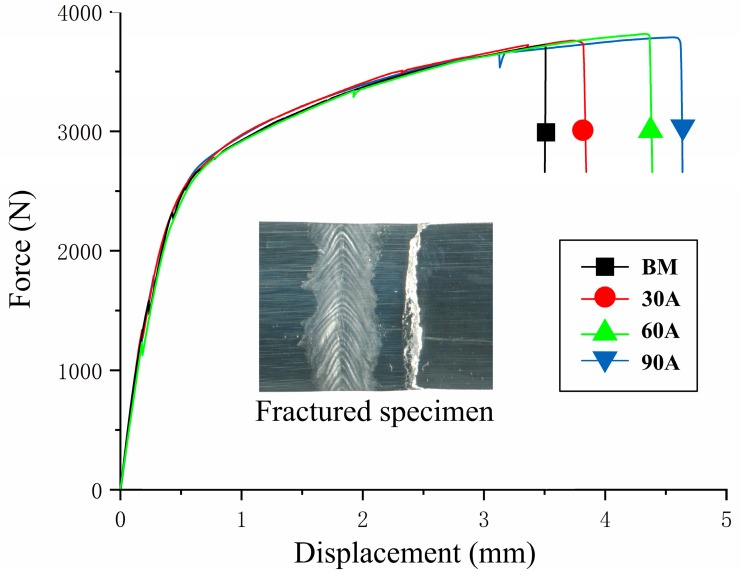
Tensile force–displacement curves of different currents.

**Figure 7 materials-12-00977-f007:**
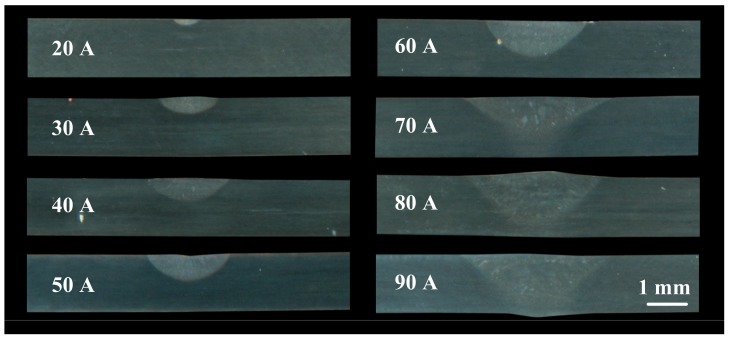
Cross-sections under different current conditions.

**Figure 8 materials-12-00977-f008:**
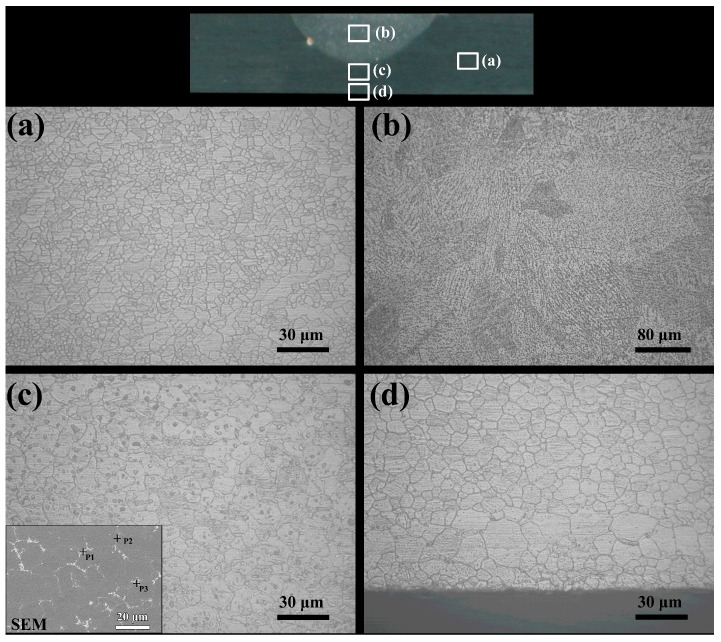
Microstructure in different regions: (**a**) BM (base metal), (**b**) FZ (fusion zone), (**c**) PMZ (partly melting zone), and (**d**) HAZ (heat-affected zone).

**Figure 9 materials-12-00977-f009:**
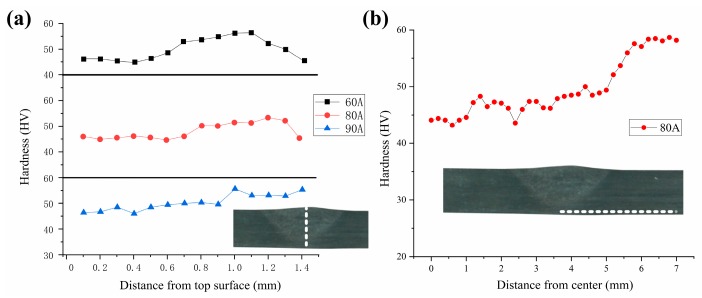
Hardness distribution of the treated area: (**a**) longitudinal line, and (**b**) transversal line.

**Figure 10 materials-12-00977-f010:**
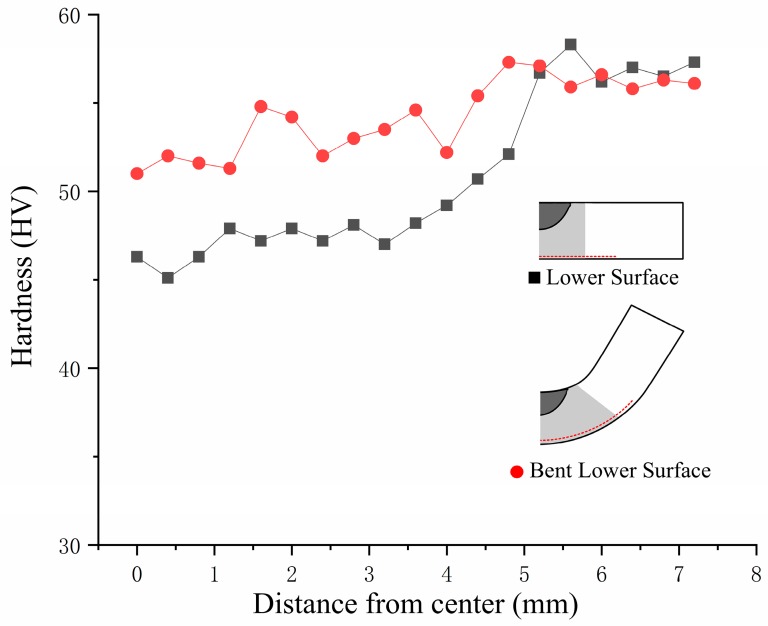
Hardness distribution of the lower surface before and after bending of the sheet.

**Figure 11 materials-12-00977-f011:**
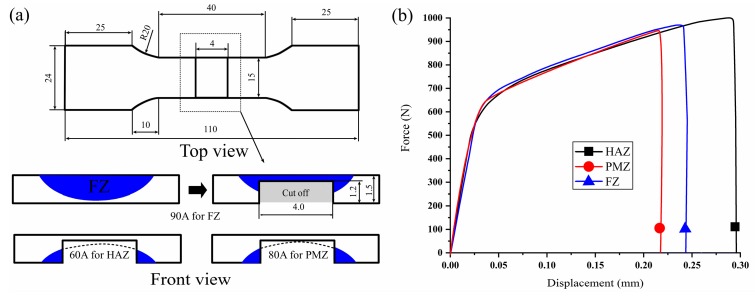
Test of the mechanical properties of different areas: (**a**) Tensile specimen schematic (**b**) Tensile test results.

**Figure 12 materials-12-00977-f012:**
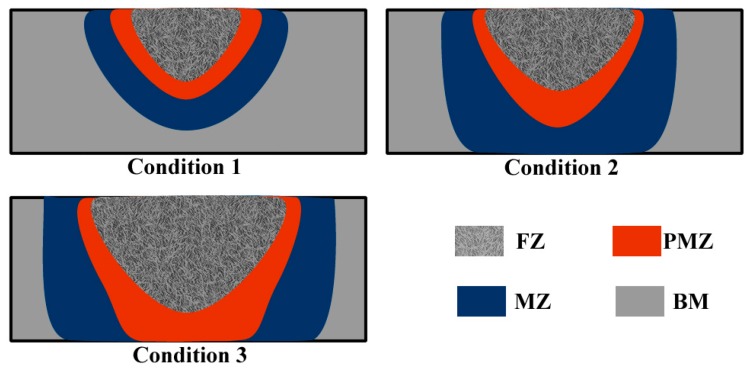
Schematic of sheet evolution under different parameters.

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
