# Peer review of "Bending Properties of Mg Alloy Tailored Arc-Heat-Treated Blanks"

_materials, 2019, doi:10.3390/ma12060977_

Round 1
Reviewer 1 Report
- The methods section miss some information. What Force did you use for hardness measurement? What was the measurement range of the load cell at the tensile and v-shape bending tests? What did you use as movement parameter while tensile testing? The speed of the traverse or something different? Please rework this section and go more into detail.
- The scale bar in Fig. 2 are for both pictures the same?
Line 119 – I understand what you want to say but it there is the comparison missing. Please change the sentence about the Al alloy for better understanding. Like “Compared with Al…”
- Which force did you use for hardness measurements?
- Could you draw a line for the hardness of base material into the three diagrams in Fig. 9? It would be nice to see how are the differences to the base material.
- How did you determined the intermetallic compound Mg17Al12? There was no picture about that. Please go more into detail here.
Author Response
Comments
- The methods section miss some information. What Force did you use for hardness measurement? What was the measurement range of the load cell at the tensile and v-shape bending tests? What did you use as movement parameter while tensile testing? The speed of the traverse or something different? Please rework this section and go more into detail.
ü The details have been added in the method section.
- The scale bar in Fig. 2 are for both pictures the same?
ü Yes, scale bar is same for both pictures.
Line 119 – I understand what you want to say but it there is the comparison missing. Please change the sentence about the Al alloy for better understanding. Like “Compared with Al…”
ü The discussion about Al alloys is somewhat inappropriate, so I have deleted the relevant sentence.
- Which force did you use for hardness measurements?
ü The microhardness tests were performed with a Vickers hardness tester with a period of 10 s and a load of 50 g.
- Could you draw a line for the hardness of base material into the three diagrams in Fig. 9? It would be nice to see how are the differences to the base material.
ü The hardness result has been added as shown in Fig.9b. Transversal hardness distribution shows that the hardness decreases from BM to HAZ, and the soften zone width by one pass reaches 10mm.
- How did you determined the intermetallic compound Mg17Al12? There was no picture about that. Please go more into detail here.
ü The microstructures were observed by SEM with EDS. According to the element spot analysis result of EDS, the phase component in grain boundaries and crystals are very close to that of Mg17Al12. Relevant results have been added in Fig. 8.

Reviewer 2 Report
1- Explain briefly the tailored heat treated blank in the abstract to clarify it for readers what you mean by this process.
2- Lines 116-121 & 158-161: Authors are describing the untreated magnesium alloy bending test with treated one and all of sudden compare this with 6061-T6 which is Al-Si-Mg alloy. What is the point of this comparison here? The authors need to make it clear for the readers the reason for this comparison between AZ31 and 6061 alloys. It would have made more sense if the arc welding had been compared with other methods such as laser welding than a different metallic system.
3- Line 126: According to figure 3a, the bending displacement of the BM is not 8.3 mm but less than 7mm. Please correct it.
4- How double-pass welding affect the V-bending properties? Would the similar results be expected?
5- Why the scale bar of figure 8 b is different than others.
Author Response
1- Explain briefly the tailored heat treated blank in the abstract to clarify it for readers what you mean by this process.
ü The details have been added in the abstract.
2- Lines 116-121 & 158-161: Authors are describing the untreated magnesium alloy bending test with treated one and all of sudden compare this with 6061-T6 which is Al-Si-Mg alloy. What is the point of this comparison here? The authors need to make it clear for the readers the reason for this comparison between AZ31 and 6061 alloys. It would have made more sense if the arc welding had been compared with other methods such as laser welding than a different metallic system.
ü The discussion about Al alloys is somewhat inappropriate, so I have deleted the relevant sentence.
3- Line 126: According to figure 3a, the bending displacement of the BM is not 8.3 mm but less than 7mm. Please correct it.
ü Thank you, and the mistake has been corrected.
4- How double-pass welding affect the V-bending properties? Would the similar results be expected?
ü In arc treating of Mg sheet, the sheet cools down quickly due to the fast-forward speed of the arc, so the temperature of first pass does not affect the second pass. Different passes do not overlap and the microstructures of first pass also does not affect the second pass. Therefore, I think similar results can be obtained in multiple pass by controlling the distance between passes.
5- Why the scale bar of figure 8 b is different than others.
ü The grains in the FZ is coarser than that in the HAZ and BM. Different amplification is used when observing different zones, so the scale bar of figure 8 b is different.
